# The Effect of Prophylactic HPV Vaccines on Oral and Oropharyngeal HPV Infection—A Systematic Review

**DOI:** 10.3390/v13071339

**Published:** 2021-07-11

**Authors:** Kristoffer Juul Nielsen, Kathrine Kronberg Jakobsen, Jakob Schmidt Jensen, Christian Grønhøj, Christian Von Buchwald

**Affiliations:** Department of Otorhinolaryngology, Head and Neck Surgery and Audiology, Rigshospitalet, University of Copenhagen, Inge Lehmanns vej. 7, 2100 Copenhagen, Denmark; kristofferjuulnielsen@hotmail.com (K.J.N.); kathrine.kronberg.jakobsen@regionh.dk (K.K.J.); jakob.schmidt.jensen.01@regionh.dk (J.S.J.); christian.groenhoej@regionh.dk (C.G.)

**Keywords:** oropharyngeal cancer, human papillomavirus, vaccines, head and neck cancer, oncology

## Abstract

Human papillomavirus (HPV) imposes an increased risk of developing cervical, anal and oropharyngeal cancer. In the Western world, HPV infection is currently the major cause of oropharyngeal cancer. The effectiveness of HPV vaccines for oral or oropharyngeal HPV infection is yet to be determined. This study conducted a systematic literature search in *Pubmed* and *Embase*. Studies investigating the impact of HPV vaccines on oral or oropharyngeal HPV infection were enrolled. This review reports the relative prevention percentage (RPP), including a risk of bias assessment as well as a quality assessment study. Nine studies were included (48,777 participants): five cross-sectional studies; one randomized community trial study (RCT); one longitudinal cohort study; and two case-control studies. A significant mean RPP of 83.9% (66.6–97.8%) was calculated from the cross-sectional studies, 82.4% in the included RCT and 83% in the longitudinal cohort study. Further, two case-control studies that measured antibody response in participants immunized with HPV vaccines were included. Respectively, 100% and 93.2% of participants developed HPV-16 Immunoglobulin G (IgG) antibodies in oral fluids post-vaccination. Analysis of the studies identified a significant decrease in vaccine-type oral or oropharyngeal HPV infections in study participants immunized with HPV vaccines across study designs and heterogenous populations. Further, a significant percentage of participants developed IgG antibodies in oral fluid post-vaccination.

## 1. Introduction

Human papillomavirus (HPV) imposes a risk of developing cervical, anal and oropharyngeal cancer. HPV infection is currently the major cause of oropharyngeal cancer in the Western world and the incidence of oral and oropharyngeal HPV infection is increasing [1,2,3,4,5,6]. Some studies suggest that the proportion of oropharyngeal squamous cell carcinoma (OPSCC) associated with HPV is 70% in North America and 73% in Europe, respectively [7].

The effect of prophylactic HPV vaccines is well documented for prevention of certain HPV types in cervical and anal cancer [8], but a prophylactic effect on oral and oropharyngeal HPV infection is yet to be established.

The prevalence of oral and oropharyngeal HPV infections varies greatly amongst countries [2,9,10], and some studies, especially from Western countries, suggest that by the end of 2020, HPV will cause more oropharyngeal cancer than cervical cancer in high-resource countries [10]. The incidence of oropharyngeal cancer is much higher for males than for females [11]. Considering that the HPV vaccination rate is significantly higher amongst females [1], primarily due to regulation and the well-established effect of HPV vaccines on cervical cancer, it is worth investigating the relationship to oral and oropharyngeal HPV infection as well. 

This review calculated/extracted a relative prevention percentage (RPP) in oral and oropharyngeal HPV infection amongst vaccinated and unvaccinated individuals as a surrogate goal for vaccine effectiveness, including studies that examine antibody response to vaccination as an alternative surrogate goal for immunization, with the aim of providing an analysis of established evidence. The aim of the study was to provide a systematic analysis and comparison of existing studies in order to provide evidence on the effect of HPV vaccines on oral and oropharyngeal HPV infection.

## 2. Materials & Methods 

This systematic review was conducted with reference to the Preferred Reporting Items for Systematic Reviews and Meta-Analyses (PRISMA) statement (Figure 1) [12].

### 2.1. Systematic Literature Search

One author (KN) systematically searched PubMed and EMBASE for articles in the English and Scandinavian languages. The search was last updated on ultimo January 2021. Studies published in the period from January 2016 to March 2021 were included. The following keywords (MeSH terms included in PubMed) were used: HPV OR Human papillomavirus OR Papillomaviridae AND Vaccine OR HPV-vaccine OR Quadrivalent vaccine OR Gardasil-9 OR Cervarix AND Oral OR Mouth OR Oropharyngeal. Inclusion criteria were studies that investigated the association between HPV vaccines and oral and/or oropharyngeal HPV infection through one or both of the following designs: (1) using a population of individuals who have received the HPV vaccine and comparing incidence of oral or oropharyngeal HPV infection to a matched population or (2) assessment of antibody response following administration of HPV vaccines. Studies with other objectives were excluded.

### 2.2. Risk of Bias Assessment in Randomized Study and Quality Assessment

For the randomized control studies, the Cochrane risk of bias tool RoB-2 [13] was used to assess bias. A total overall bias of the article was determined using domains and ‘signaling questions’ provided by the RoB-2 tool. A summary of the assessment is provided in Figure 2 and the complete analysis can be found in Appendix A: ‘ROB 2 Appendix A.xlsm.’ For the non-randomized studies, study quality was assessed using the ‘study quality assessment tool’ from the NIH [14]. Evaluation questions and results from the assessment are presented in Appendix A.

## 3. Results

### 3.1. Study Characteristics 

The literature search generated 1530 studies of which we included five cross-sectional studies [2,9,10,15,16], one randomized community trial study [17], one longitudinal cohort study [18] and two case-control studies [10,19], which in total encompasses 48,777 participants. A full overview of study characteristics can be found in Table 1.

Four cross-sectional studies [2,10,15,16] observed a significant reduction in the presence of oral and oropharyngeal HPV infection after immunization with HPV vaccines. One cross-sectional study [9] proved inconclusive regarding vaccine effectiveness on oral and oropharyngeal HPV infection due to only four participants testing positive for any HPV type in oral samples, and was subsequently not included in the calculation of the mean RPP. The four cross-sectional studies had an RPP of 82.2% [2], 72.0% [10], 89.8% [15] and 91.7% [16], corresponding to a mean RPP of 83.9%. 

The enrolled studies used different methods for collecting data and biological material from participants. One study [2] collected patient data from the National Health and Nutrition Examination Survey (NHANES), while others relied on self-reported vaccination status [10,15,16]. Biological material was collected in various ways; most studies relied on a swap-test from the oral cavity and tonsils [9,11,15,20]. Other studies used a gargle/rinse sample [2,10] while one study examined tissue from the tonsils of patients undergoing tonsillectomies [15].

One cross-sectional study [9] (*n* = 312) found no correlation between HPV prevalence and vaccination status. The authors of this study claimed that results regarding oral and oropharyngeal infection proved inconclusive due to only four participants testing positive for any type of oral and oropharyngeal HPV infection.

The community randomized trial [17] (*n* = 38,621) examined the effectiveness of the AS04-adjuvanted HPV-16/18 vaccine in reducing oral HPV infection in young females. Three arms of eleven communities (totaling 38,631 participants) were enrolled and participants were blinded regarding vaccine allocation. The trial showed a significant reduction in oral HPV-16/18 and found an RPP of 82.4% in HPV-16/18 six years post-vaccination.

The two case-control studies [11,20] (*n* = 34 and *n* = 150) investigated antibody response in oral and oropharyngeal mucosal fluids and sera pre- and post-immunization with HPV vaccines. Both studies showed that a significant percentage of the population developed a vaccine-specific antibody response post-immunization. Sera and saliva were collected using mouthwash and Merocel sponges at both day one and month seven, and were tested for anti-HPV 16/18 immunoglobulin G (IgG) levels by an L1 virus-like particle-based enzyme-linked immunosorbent assay. In one [11] study 100% of men (*n* = 150) seroconverted and the majority of participants developed anti-HPV16/18 antibodies in the oral cavity following vaccination: 96% for anti-HPV 16 and 72% for anti-HPV 18. One study also showed that in vitro antibodies were able to neutralize HPV pseudovirions [20].

One longitudinal cohort study [18] assessed the risk of oral and oropharyngeal HPV infections amongst sexually active female adolescents who received the quadrivalent HPV vaccine. This study analyzed the repeated collection of oral rinse specimens from participants in healthcare clinics. HPV DNA was analyzed using PCR. The study found a significant reduction (83%) in the prevalence of oral and oropharyngeal HPV infection in vaccinated individuals vs. unvaccinated individuals.

### 3.2. Risk of Bias Assessment and Study Quality Assessment

A Cochrane risk of bias assessment was performed on the enrolled randomized study [17] using the RoB-2 tool. The overall assessment of bias was classified by author K.J.N and the RoB-2 algorithm as ‘some concerns.’ A summary of the assessment is presented in Figure 2 and the full analysis can be found in Appendix A. For the non-randomized studies, the quality assessment tool from the NIH was used to evaluate study quality based on 14 predetermined questions for the observational cohort and cross-sectional studies as well as 12 questions for the case-control studies. Each individual study received a final overall score. The assessments can be found in Appendix A.

### 3.3. HPV-Specific Antibodies Pre- and Post-Vaccination

In the aforementioned case-control studies, neutralizing HPV-targeted serum and saliva antibodies were collected from participants before and after immunization. In one study [20], 100% of participants (*n* = 34) seroconverted and developed IgG antibodies in oral fluids against vaccine-targeted HPV, in this case HPV 6, 16 and 18, post-vaccine. In vitro testing showed that antibodies were capable of neutralizing HPV pseudovirions. In the other case-control study [11], similar procedures were used and the nine-valent HPV vaccine was administered. One hundred percent of participants (*n* = 150) seroconverted and 93.2% developed anti HPV-16 antibodies while 72.1% developed anti HPV-18 antibodies within seven months. After seven months, antibodies in oral fluids were significantly correlated to serum levels.

## 4. Discussion

Overall, the comparative studies showed a high HPV vaccine effectiveness on oral and oropharyngeal HPV infection regardless of study design. The cross-sectional studies had a mean relative reduction percentage of 83.9%, the RCT [17] had a relative reduction percentage of 82.4% and the longitudinal cohort study calculated an RPP of 82.0%. In total, the average RPP amongst all the studies was 82.7% (CI 81.8–83.7%.) The similarity in positive outcome across study types suggests a significant and stable effectiveness of HPV vaccines on vaccine-type oral and oropharyngeal HPV infection. Similarly, the case-control studies used measurements of antibody response post-vaccination as an indicator of the effectiveness of HPV vaccines on oral and oropharyngeal HPV infection. The fact that antibodies proved to neutralize HPV pseudovirions in vitro generates exciting prospects of longer studies with larger populations to examine the long-term immunization of HPV vaccines on oral and oropharyngeal HPV infection. A causal relationship between neutralizing antibodies and immunization against oral and oropharyngeal HPV infection is yet to be established.

In Denmark, the HPV vaccine is offered to girls and boys between the ages of 12 and 18 years and is a part of the tax-funded vaccination program. Until ultimo 2021, men who had sex with men between the ages of 18 and 26 years could elect for a tax-funded vaccine as well. [21] In the Netherlands, as of 2021, the National Institute for Public Health (RIVM) provides HPV vaccines for both genders at the age of nine as part of the national vaccination program. Currently, in Denmark and other nations, HPV vaccines are administered on the indication of precursor lesions in cervical, vulva, vaginal and anal cancer, but not oropharyngeal cancer [21]. The rationale underlying this is that in cervical, vulva, vaginal and anal cancers premalignant precursor lesions make it possible to prove the effectiveness of HPV vaccines on said lesions. The same causal relationship has naturally not been established in oropharyngeal cancer, as no premalignant lesions are known. The results of this systematic review suggest that due to high vaccine effectiveness on oral and oropharyngeal HPV infection, the introduction of a pan-gender vaccination program would constitute a significant decrease in oropharyngeal cancers globally. Further, long-term randomized studies could provide a causal relationship regarding immunization with HPV vaccine and a decrease in incidence of HPV positive oropharyngeal cancers. Such a study is, however, a difficult task, as a randomized study would require the primary endpoint to be the contraction of oropharyngeal cancer, which would not be ethically sound. Another way to establish further evidence would be, if possible, to identify a thus far unknown oropharyngeal precursor lesion. This would make it possible to create an ethically sound randomized study where vaccine effectiveness on oropharyngeal precursor lesions could be examined.

It is worth considering that a persistent oral or oropharyngeal HPV infection is rare in the general population, even amongst risk population e.g., partners of cervical HPV positive women [19]. However, these risk populations constitute a significantly higher risk of testing positive for oral or oropharyngeal HPV infection when compared to the general population [22]. In this regard, a probable uncertainty is the fact that testing for HPV e.g., in patients with acute pharyngitis, which might reflect acute oropharyngeal HPV infection, is rare. This is because there is no therapeutic gain in diagnosing HPV in acute oropharyngeal pharyngitis.

No standardized procedures have been decided on for the collection of oral and oropharyngeal samples regarding determination of HPV infections. This has caused inconsistencies in the methods used to obtain oral and oropharyngeal samples. In the included studies, the following methods were used: Floq Swab; [9]; oral rinse/gargle sample [2]; unspecific oropharyngeal sample [17]; biological rinse sample [10]; Oracol S10 devices [15]; Orcellex brushes [15]; OraSure pads [20]; Merocel sponges [11]; and a rinse and alcohol gargle sample [18]. These inconsistencies in sample collection could theoretically cause discrepancies in the analysis of oral and oropharyngeal HPV, which could potentially cause inadequate DNA extraction from the samples. One study [23] suggests that a ‘rinse’ method is more accurate than a ‘swab’ method in the extraction of HPV DNA from oral samples. It is proven that the detection of HPV in patients with macroscopically visible tumors is more sensitive than in patients where the tumor is not visible. One study concludes that all “superficial” HPV detection methods are insufficient in collecting material when the tumor is not macroscopically visible [24]. This calls for further studies to establish the most precise method and to standardize sample collection in order to ensure equal and adequate HPV DNA extraction in future studies.

This systematic review has several strengths. Studies with different designs were enrolled, i.e., RCT, cross-sectional, case-control and cohort studies, in order to undertake a broad comparative analysis with the same predetermined outcome-specific assessment. In the same way, studies with different methods of assessing incidence and prevalence of oral and oropharyngeal HPV infection participated in the analysis. The relevant data was extracted and the RPP of each study was calculated in order to perform a diverse comparative analysis. A similarity in results across different methods supports the legitimacy of the outcome. Furthermore, the studies consisted of heterogenic populations, making it possible to determine the general effectiveness of HPV vaccines on oral and oropharyngeal HPV infections.

One limitation of this study is that the averages calculated in the RPP of the cross-sectional studies are not weighted averages. This is due to some of the studies not having full transparency regarding which participants had oral and oropharyngeal HPV infection as opposed to any HPV infection, thereby making it difficult to calculate a weighted average. Another consideration is the inconsistent sample techniques as mentioned above.

A further consideration is the inability to distinguish between samples from the oral cavity and from the oropharynx. This results in uncertainty and makes it difficult isolating vaccine effectiveness on oropharyngeal HPV infection. In future studies, one way of isolating oropharyngeal material would be to biopsy tissue from the tongue basis and tonsil and analyze it for HPV infection, which only one of the studies did [15]. Regarding the epidemiology of oropharyngeal HPV infection, it can be difficult to quantify the prevalence in a certain population. This is due to the microanatomic location of HPV, which is presumed to reside in the base crypts of the tonsils and the depths of the tongue basis [25]. It is possible, that the location of HPV could cause the infection to go undetected, thereby making the immediate prevalence appear lower than the actual prevalence, which creates an uncertainty and must be considered a limitation.

The RPP of HPV vaccines (bivalent, quadrivalent and nine-valent) on oral and oropharyngeal infection proved significant, though it is still relevant to discuss why the RPP is not closer to 100%. One reason for this could be human error or inadequate technique in sample collection. Another explanation could be recall bias in the studies where vaccine status was self-reported. A third explanation could be participants having contracted oral or oropharyngeal HPV infection prior to the vaccination. In many communities, particularly religious communities, there is a taboo surrounding premarital sexual relations, which in theory could inhibit participants from truthfully reporting sexual status prior to vaccination trials. A final explanation could be some participants not developing adequate antibody response post-immunization. To investigate this hypothesis, further studies are needed.

## 5. Conclusions

A significant decrease in vaccine-type oral and oropharyngeal HPV infection in study participants immunized with HPV vaccines was calculated or extracted, across study design and heterogenous populations. Similarly, a significant percentage of participants in the case-control studies developed IgG antibodies in the oral cavity post-immunizations with HPV vaccines, which presents an alternative surrogate goal for vaccine effectiveness. Furthermore, the similarity in RPP across study designs supports the legitimacy of the results.

## Figures and Tables

**Figure 1 viruses-13-01339-f001:**
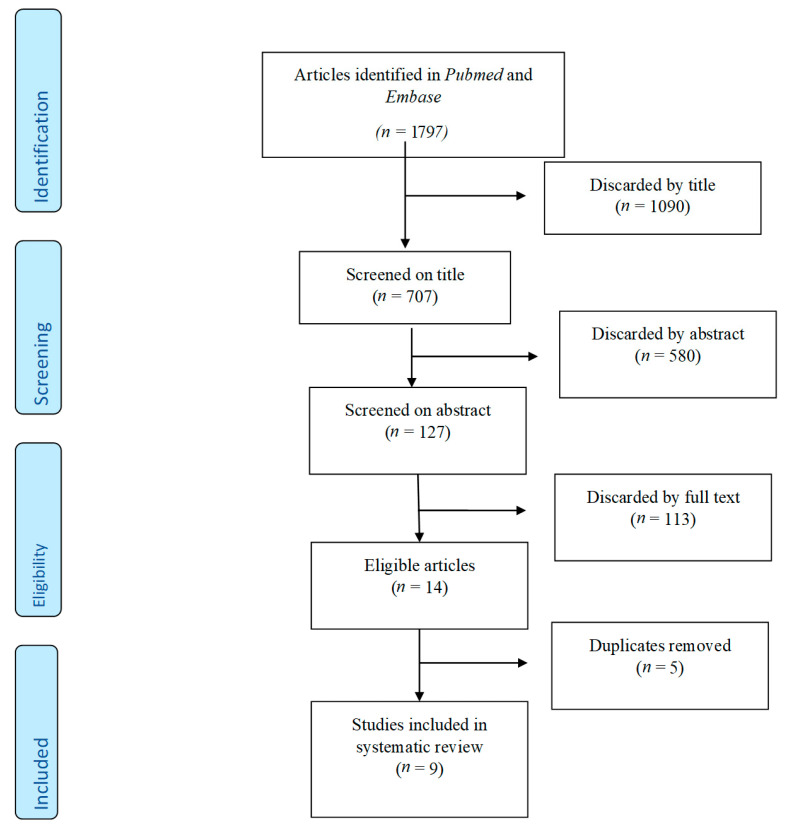
PRISMA flowchart.

**Figure 2 viruses-13-01339-f002:**
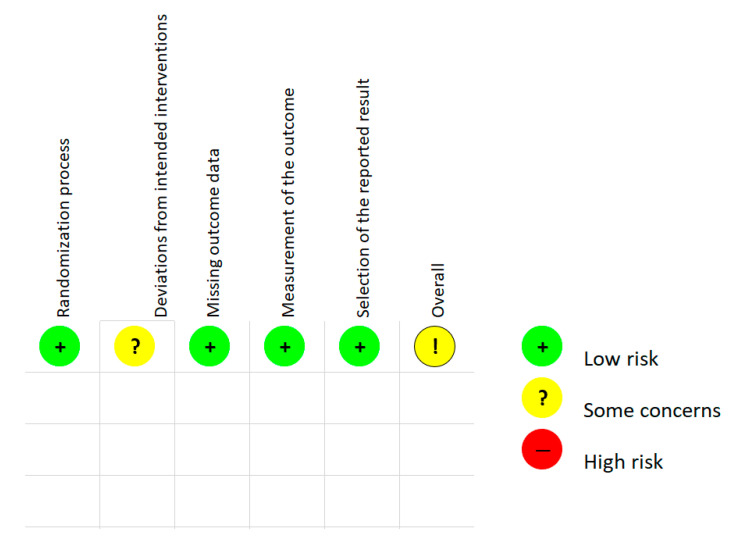
RoB-2 assessment. (Lethinen et al. 2019).

**Table 1 viruses-13-01339-t001:** Study characteristics.

Study	Study Design	Participants	Key Findings	Odds/Risk Ratio/Prevalence Ratio	*p*-Value	Method
Espen Enerly [9] 2019	Observational study- cross sectional study	312	In total four oral samples were positive for any type of HPV, and all of these participants had received at least one HPV vaccine before oral sexual debut. Results for infections other than oral HPV infection are not included in the article.	N/A * Results of vaccine effect on oral HPV infection are disregarded due to small sample size.	%	Facebook advertisement to recruit participants. Self-sampled oral and vaginal specimens using Evalyn brush and FLOQ-swab. Sexual behavior ascertained through questionnaire.
Anil K. Chaturvedi [2] 2017	Cross-sectional study	2627	The prevalence of oral HPV (16/18/6/11) infections was significantly reduced in vaccinated vs. unvaccinated individuals. This corresponds to an estimated 82.2% (5.7%–98.5%) reduction on the prevalence after adjustments were made.	0.11% of vaccinated vs. 1.61% of unvaccinated individuals had an oral HPV infection.	Adjusted *p*-value = 0.008	Patient data was collected from NHANES (2011–2014). Vaccination status was self-reported.
Matti Lehtinen [17] 2019	Community randomized trial	38,631	Vaccine effectiveness on HPV (16/18, 31/45 and 31/33/45) was, respectively, 82.4% CI: (47.3–94.1); 75.3% CI: (12.7–93.0); and 69.9% CI: (29.6–87.1). The AS04 vaccine showed effectiveness on HPV infections in adolescent females up to 6 years post-vaccination.	Relative reduction in HPV16/18: 82.4%	%	Three arms of 11 communities were enrolled and compared. Participants were blinded to vaccine allocation. HPV DNA prevalence was determined by SPF-10 LiPA and Multiplex type-specific PCR.
Andres Castillo [10] 2019	Cross-sectional study	1784	HPV vaccination was associated with the reduction of HPV-16 exposure percentages in the oral and oropharyngeal cavity. 72% reduction in HPV-16 detection in students immunized with two doses.	ODDS RATIO HPV16 in vaccinated vs. unvaccinated: 0.28 (95% CI: 0.07–0.88).	*p* = 0.01; calculated using regression model.	HPV-16 DNA was detected in samples from the oral cavity and throat of 1784 high school students of both genders, aged 14–17 years old. The number of vaccinated girls was 944 vs. 95 unvaccinated girls and 745 unvaccinated boys.
Hisham Mehanna [15] 2019	Cross-sectional study	940	Overall, oropharyngeal HPV-16 prevalence was significantly lower in vaccinated vs. unvaccinated females. In contrast, prevalence of any oropharyngeal HPV type was similar in vaccinated and unvaccinated females. Oropharyngeal HPV-16 prevalence in unvaccinated males was similar to vaccinated females.	HPV-16 Prevalence in vaccinated vs. unvaccinated women: 0.5% vs. 5.6%. Prevalence in unvaccinated males was similar to vaccinated females (0% vs 0.5%, *p* > 0.99).	*p* = 0.04 *p* > 0.99	Subjects aged 0–65 years undergoing tonsillectomy for nonmalignant indications were recruited in six hospitals in the United Kingdom. Vaccination status obtained from health authorities. All samples were centrally tested for HPV DNA by polymerase chain reaction.
A Handisurya [20] 2019	Case-control study	34	HPV vaccination induced type-specific antibody response in oral fluid and sera. In vitro the antibodies in oral fluid were capable of neutralizing HPV pseudovirions indicating protection from infection.	N/A	*p* < 0.001	Oral fluid and sera were collected from females before and after administration of the quadrivalent vaccine. IgG and neutralizing antibodies of HPV 6 and 16/18 were analyzed pre- and post-vaccination and compared to unvaccinated females.
Jacqueline M. Hirth [16] 2017	Cross-sectional study	3040	Lower prevalence of oral HPV types in vaccinated vs. unvaccinated individuals. Prevalence was the same amongst participants on non-vaccine type oral HPV infection.	HPV16 vaccinated vs. unvaccinated prevalence: 0.09 vs. 0.84. HPV18 0.07 vs. 0.29	16: 0.01; 18: 0.15; Any high risk: 0.04.	Cross-sectional data obtained from NHANES. Participants provided oral samples and questionnaires were used to ascertain vaccination status amongst participants.
Ligia A. Pinto [11] 2016	Case-control study	150	100% of men seroconverted, and the majority of individuals developed detectable anti-HPV-16 and anti-HPV-18 antibodies (up to 96% and 72%, respectively) at the oral cavity following vaccination.	%	%	Sera and saliva collected in mouthwash and Merocel sponges at day 1 and month 7 were obtained from 150 men who received Gardasil at day 1 and months 2 and 6. Specimens were tested for anti-HPV-16 and anti-HPV-18 IgG levels by an L1 virus-like particle-based enzyme-linked immunosorbent assay.
Nicolas F. Schlecht [18] 2019	Longitudinal cohort study	1259	Large risk of oral HPV-infection in unvaccinated women. Vaccination associated with significant decrease in prevalence of oral HPV-infection. (83% decrease in risk of oral HPV infection in vaccinated individuals.)	Odds ratio: 0.17 CI: (0.04–0.998).	%	Repeated collection of oral rinse specimens from sexually active female adolescents in healthcare clinics. Included vaccinated and unvaccinated individuals. HPV-DNA was analyzed using PCR.

* N/A: data could not be extracted or calculated from study.

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
