# Peer review of "The Effect of Prophylactic HPV Vaccines on Oral and Oropharyngeal HPV Infection—A Systematic Review"

_viruses, 2021, doi:10.3390/v13071339_

Round 1

Reviewer 1 Report

The authors conducted a meta-analysis on studies that tested for prophylactic effects of HPV-vaccinations against oropharyngeal infections with high risk HPV subtypes. 9 studies were included. Lower prevalences of HPV were found in vaccinated study populations. In vaccinated participants, most had HPV-specific antibodies after vaccination in saliva and serum. The authors conclude that HPV vaccination should be considered in both sexes to prevent oropharyngeal HPV infection and likely HPV-associated OPSCC.

Major:

  1. The introduction would benefit from more details about epidemiologic numbers on oropharyngeal HPV infection which is often tricky to diagnose or can be missed easily when infection is "healed" because HPV propbably resides in the base of tonsil crypts or in the depth of base of tongue which should be mentioned in the introduction and/or should be discussed.
  2. Testing for HPV, e.g. in acute pharyngitis, which might reflect acute oropharyngeal HPV infection, is rare because there is no therapeutic consequence as the authors discuss.
  3. Persistent oropharyngeal infection of HPV at least seems to be rare, even in risk populations (compare e.g. Eggersmann et al 2019, Cossellu et al 2018). Regarding the techniques of obtaining HPV from oral cavity/oropharynx, at least partially studies of Barbara Kofler (3 English studies on HPV infection/HPV-associated OPSCC in Cancers) and again Eggersmann et al 2020 should be discussed

Minor: Figure 2 seems to be cut at the upper edge.

Overall, I recommend revision of the aforementioned points to strengthen introduction and discussion of the manuscript. Aside, this review performs an important meta-analysis on HPV vaccination in the context of oropharyngeal HPV infection.

Author Response

Point 1:

The introduction would benefit from more details about epidemiologic numbers on oropharyngeal HPV infection which is often tricky to diagnose or can be missed easily when infection is "healed" because HPV propbably resides in the base of tonsil crypts or in the depth of base of tongue which should be mentioned in the introduction and/or should be discussed.

 Response 1:

Very valid point. We have added a short paragraph on the prevalence of Oropharyngeal HPV infection in North America and Europe citing “Palmer et. Al.” in the introduction starting at line 32:

 “Some studies suggests that the proportion of oropharyngeal squamous cell carcinoma (OPSCC) associated with HPV is respectively 70% in North America and 73% in Eu-rope [7].”

 In regards to the tricky diagnosis of the infection we have added a paragraph starting at line 245:

 “In regards to the epidemiology of oropharyngeal HPV infection it can be difficult to quantify the prevalence in a certain population. This is due to the microanatomic loca-tion of HPV which is presumed to reside in the base crypts of the tonsils and the depths of the tongue basis [22]. It is possible, that the location of HPV could cause the infection to go undetected, thereby making the immediate prevalence appear lower than the actual prevalence, which creates an uncertainty and must be considered a limitation.”

Point 2:

Testing for HPV, e.g. in acute pharyngitis, which might reflect acute oropharyngeal HPV infection, is rare because there is no therapeutic consequence as the authors discuss.

Response 2:

 This is a very good point, I have added a short paragraph starting at line 204:

in this regard, a probable uncertainty is the fact that testing for HPV fx. In patients with acute pharyngitis, which might reflect acute oropharyngeal HPV infection, is rare. This is because there is of no therapeutic gain in diagnosing HPV in acute oropharyngeal  pharyngitis.

Point 3:

Persistent oropharyngeal infection of HPV at least seems to be rare, even in risk populations (compare e.g. Eggersmann et al 2019, Cossellu et al 2018). Regarding the techniques of obtaining HPV from oral cavity/oropharynx, at least partially studies of Barbara Kofler (3 English studies on HPV infection/HPV-associated OPSCC in Cancers) and again Eggersmann et al 2020 should be discussed.

Response 3:

Thank you for this very relevant point and the references. We have added a short paragraph on the prevalence of persistent oral/oropharyngeal infection in the general population compared to a risk population e.g. partners of cervical HPV positive women starting at line 202

 “It is worth considering that a persistent oral or oropharyngeal HPV infection is rare in the general population, even amongst risk population e.g. partners of cervical HPV positive women [22]. However, these risk populations constitute a significantly higher risk of testing positive for oral or oropharyngeal HPV infection when compared to the general population [23].”

 Regarding the techniques of detection of HPV, we appreciate the suggested references and have compared them in the discussion starting at line 216:

 “Another study concludes that a ‘brushing’ technique has sufficient sensitivity in detecting superficial HPV in oropharyngeal infection [25]. It is proven that the detection of HPV in patients with macroscopically visible tumors is more sensitive than in patients where tumor is not visible. One study concludes that all “superficial” HPV de-tection methods are insufficient in collecting material when tumor is not macroscopically visible.”

 Minor comments:

Figure 2 seems to be cut at the upper edge.

 Response to minor comments:

 We do not see a cut off in the figure. Could it be in the conversion of the file type? E.g. from word to pdf?

Reviewer 2 Report

A very timely and well-written systematic review which should be published.

Author Response

Thank you very much for the encouraging words on our study. We are very excitited on the prospect of publishing our study in Viruses 

Round 2

Reviewer 1 Report

After revision, the manuscript is strengthened. Except for the following minor points that should be changed in a minor revision, I can recommend accepting the manuscript for publication.

  1. In figure 2, it still says "Deviations from intended interve" in the PDF, I guess, is should say "intervention".
  2. Reference 21 in the reference list says "ssi.pdf."
  3. In the added line page 8 line 205 it says "fx. In". I would recommend changing it to "e.g. in".
  4. In the revision, there was "Another study concludes that a ‘brushing’ technique has sufficient sensitivity in detecting superficial HPV in oropharyngeal infection [25]." added which is misleading. In the cited reference, HPV testing was used as a tumor marker for HPV-related OPSCC recurrence, not infection. This should be corrected.
  5. For the following added part, the reference should be cited (I guess, Eggersmann et al. 2020?!): "It is proven that the detection of HPV in patients with macroscopically visible tumors is more sensitive than in patients where tumor is not visible. One study concludes that all “superficial” HPV detection methods are insufficient in collecting material when tumor is not macroscopically visible."

Author Response

Response to Reviewer 1 Comments

Point 1:

In figure 2, it still says "Deviations from intended interve" in the PDF, I guess, is should say "intervention".

 Response 1:

Yes, we see it now and have corrected the figure in the manuscript. Thank you for pointing this out.

Point 2:

Reference 21 in the reference list says "ssi.pdf."

Response 2:

This is a mistake and has been corrected in the manuscript. We were referencing the Danish children vaccination program and have now found the proper reference. Thank you.

Point 3:

In the added line page 8 line 205 it says "fx. In". I would recommend changing it to "e.g. in".

Response 3:

This has been corrected in the manuscript, thank you.

 Point 4:

In the revision, there was "Another study concludes that a ‘brushing’ technique has sufficient sensitivity in detecting superficial HPV in oropharyngeal infection [25]." added which is misleading. In the cited reference, HPV testing was used as a tumor marker for HPV-related OPSCC recurrence, not infection. This should be corrected.

 We have re-read the study and decided to remove the sentence and reference from the manuscript. Thank you.

 Point 5:

For the following added part, the reference should be cited (I guess, Eggersmann et al. 2020?!): "It is proven that the detection of HPV in patients with macroscopically visible tumors is more sensitive than in patients where tumor is not visible. One study concludes that all “superficial” HPV detection methods are insufficient in collecting material when tumor is not macroscopically visible."

Response 5:

Yes indeed it should. We have cited Eggersmann manually in the manuscript, as the reference manager in Mendeley has the wrong reference. Thank you.